

# Assessing Chinese anatomists' perceptions and attitudes toward blended learning through faculty development training programs

Xin Cheng[1], Jian Bai[2,3], San-Qiang Pan[4], Yun-Qing Li[5] and Xuesong Yang[1,6]

[1] Department of Histology and Embryology, Key Laboratory for Regenerative Medicine of the Ministry of Education, Medical College, Jinan University, Guangzhou, Guangdong, China
[2] Medical College, Jinan University, Guangzhou, Guangdong, China
[3] School of Education, South China Normal University, Guangzhou, China
[4] Department of Anatomy, Medical College, Jinan University, Guangzhou, China
[5] Department of Anatomy, Histology and Embryology, K.K. Leung Brain Research Centre, The Fourth Military Medical University, Xi'an, Shanxi, China
[6] Clinical Research Center, Clifford Hospital, Guangzhou, People's Republic of China

Corresponding authors
Yun-Qing Li, yunqing@fmmu.edu.cn
Xuesong Yang,
yang_xuesong@126.com

## ABSTRACT

**Background.** As a response to the COVID-19 pandemic, the faculty development program has partially shifted to online formats over the past two years, with a specific focus on professional training related to blended learning. The effectiveness of this training is closely tied to the perceptions and acceptability of blended learning among the trainees. This study aims to evaluate the perspectives of educators on blended learning, thereby assessing the efficacy of faculty training programs.

**Methods.** Anatomical teachers were chosen as a representative sample due to their significant presence among medical science educators. Chinese anatomists were invited to participate in a survey that gauges their attitudes and readiness for blended learning.

**Results.** A total of 297 responses were collected, covering all provinces in mainland China. The findings from the survey demonstrate that Chinese anatomists hold learning flexibility in the highest regard among the various facets of blended learning. Meanwhile, the presence of a connected learning community emerged as a pivotal factor influencing anatomists' perceptions, explaining 14.77% of the total variance. Further analysis showed noteworthy disparities in anatomists' attitudes toward blending learning based on their job titles, mentorship guidance, and support from in-service institutions. Notably, lecturers showed a more pronounced engagement in the connected learning community than teachers with different job titles. Additionally, anatomists who received stronger institutional support showed higher proficiencies in learning management.

**Conclusion.** This survey revealed that Chinese anatomists attribute considerable value to aspects such as learning flexibility, a connected learning community, and effective learning management within the domain of online/blended learning. Positive attitudes toward blended learning are likely to be nurtured by mentorship and institutional support, subsequently correlating with improved training outcomes. The distinctive characteristics observed among Chinese anatomists in the context of blended learning offers insights to enhance the effectiveness of faculty training programs, thereby facilitating the evolution of future teaching strategies.

## INTRODUCTION

Medical education has undergone a gradual transformation, establishing itself as a distinctive and autonomous discipline. In the past, the prevailing assumption was that proficient healthcare professionals could effectively engage in teaching after specializing in their respective fields and joining academic institutions. However, the implementation of faculty development programs has unveiled considerable potential in enhancing the pedagogical effectiveness of medical educators, thereby yielding improved learning outcomes for students. These comprehensive faculty development initiatives aim to equip participants with the necessary knowledge, skills, and competencies to excel as teachers, leaders, and researchers. Over the past decade, the critical significance of faculty development has gained widespread acknowledgement, prompting numerous medical schools and affiliated hospitals to establish sustainable training frameworks that facilitate the systematic design and implementation of tailored faculty development programs (*Bligh, 2005*; *Hueppchen et al., 2011*; *McLean, Cilliers & Van Wyk, 2008*). Despite these endeavors, faculty development remains a complex undertaking that requires diverse forms of support, including visionary institutional leadership, appropriate resource allocation, and recognition for teaching efforts (*McLean, Cilliers & Van Wyk, 2008*).

Faculty development represents an effective approach to enhancing the expertise of faculty members across various disciplines. Within authentic educational environments, faculty development encompasses the cultivation of innovative teaching approaches, refining course structures, strengthening teacher-student relationships, and acknowledging and rewarding exceptional teaching performance (*Trowbridge et al., 2011*). In this particular context, "faculty development" primarily refers to the concept of "instructional development" (*Camblin & Steger, 2000*). The core objectives of faculty development in professional health education are to enhance faculty competency within their respective roles, such as patient care for clinicians, healthcare management for administrators, and teaching effectiveness for medical educators (*Camblin & Steger, 2000*; *Steinert, Nasmith & Daigle, 2003*; *Brinkley-Etzkorn, 2018*). This effectiveness has been consistently reaffirmed through repeated explorations of educators' viewpoints on professional training (*Avramidis & Norwich, 2002*; *Van Aalderen-Smeets & Walma van der Molen, 2015*). Consequently, faculty development has progressively become an integral part of the medical education framework. Almost all medical schools provide a diverse range of flexible faculty training programs, aiming to foster an environment that vigorously promotes faculty teaching excellence to the ultimate benefit of medical students (*Burgess et al., 2019*; *Crown, Fuentes & Freeman, 2011*; *Pan et al., 2020*).

Regrettably, the unforeseen COVID-19 pandemic posed a challenge. The established faculty development work arrangements were abruptly disturbed (*Buckley, 2020*; *Eltayar et al., 2020*; *Kachra & Ma, 2020*). During the past two years, university teaching staff

worldwide have endeavored to adapt to the new situation: the implementation of online-only education or blended online and offline learning for a specific period to sustain normal teaching activities as much as possible (*Cheng et al., 2021*). The demand for training in effectively delivering online or blended courses increased significantly. During the COVID-19 pandemic, faculty development programs augmented the capabilities of university faculty to deliver effective online teaching through diverse digital strategies (*Swaminathan et al., 2021*).

Amidst this backdrop, a crucial inquiry emerges: what is the most effective form of professional training for teachers in online/blended learning? Before addressing this issue, it is worth examining the acceptance of blended learning among teachers, as it directly influences the effectiveness of the training. Despite the frequent use of the term "blended learning", there remains ambiguity regarding its precise definition. While there is a general consensus that blended learning is the combination of face-to-face and online instruction or learning (*Hrastinski, 2019*), the teaching and learning aspects are often overlooked by blended learning researchers. In university education, educators should embrace the dual roles of teacher and learner, gaining comprehensive experience in both facets of blended learning. To illustrate, anatomists represent a significant population of medical educators and can serve as exemplars. Anatomical educators consistently strive to improve their instructional methods. However, due to the reliance on hands-on experience and dissection in anatomy, concerns about learning outcomes have arisen with online learning, impeding the progress of blended learning prior to the COVID-19 epidemic (*Harmon et al., 2021*). Additionally, Chinese anatomy education, like that in other countries, confronts numerous challenges, including reduced course hours, a scarcity of donated bodies, and an increasingly imbalanced teacher-student ratio (*Pan et al., 2020*). To effectively address these issues, blended learning or online courses offer viable solutions by providing unrestricted access to learning resources.

In the current educational landscape, a pivotal consideration is the effective and practical professional training of faculty members, enabling the smooth transition from traditional education to blended learning. Grounded in the inherent link between cognition and behaviors, the effectiveness of teacher training is contingent upon the perceptions of academic staff regarding blended learning. This study focused on Chinese anatomy educators as the target cohort, aiming to delve into their perspectives on blended learning, while also investigating potential disparities in these perspectives based on factors such as gender, age, years of teaching experience, and institutional support. Through a comprehensive analysis of survey responses from anatomical educators, this study contributes to the broader discourse on enhancing the effectiveness of blended anatomy education delivery.

## MATERIALS AND METHODS

### Survey participants and context

The survey instrument was designed to collect information on the attitudes of Chinese anatomy educators toward blended learning. The survey encompassed several dimensions,

including respondents' demographic information, their experience as anatomical educators, their professional training encompassing teaching and research roles, and their readiness toward blended learning. These questions were adapted from literature and slightly modified (*Tang & Chaw, 2013*). A Likert scale, featuring a range of 1 to 6, was employed to rate responses, with options ranging from "1 =strongly disagree" to "6 =strongly agree". The questionnaire contained 59 questions and required approximately 8 to 10 min to complete. The questionnaire was formulated in Chinese (see Appendix S1 for the English translation). A pilot survey involving seven faculty members from the institution of the first author ensured the questionnaire's clarity, which led to subsequent revision based on the feedback received.

A convenience sampling approach was used within the anatomy departments of mainland China. A considerable proportion of directors from anatomy departments in mainland China's medical schools belonged to a WeChat application messaging group ($n = 500$), a popular social media mobile platform (Tencent Holdings Ltd., Shenzhen, China) (*Gan & Wang, 2015*). The survey invitations were delivered through a group-level link *via* SoJump (Ranxing, Changsha, China) online platform. Participation in the survey was voluntary. The survey was conducted in October 2021, with the questionnaire link remaining active during the four-day span of the 36th Annual Academic Conference of the Chinese Society for Anatomy Sciences (CSAS)—the national organization for anatomists. The study was conducted with approval of the Research Ethics Committee of Jinan University (No. JNUKY-2021-038). A consent form and the questionnaire invitation link were sent to the participants simultaneously. The recipients would be defaulted as giving consent to the study and answering the survey.

## Data analysis

All statistical analyses were performed using the SPSS statistical package version 26.0 (IBM Corp., Armonk, NY, USA). The data obtained from the questionnaires were analyzed using Cronbach's $\alpha$ test to determine the internal consistency of the responses. Exploratory factor analyses were employed to identify factors that reflected the respondents' attitudes toward blended learning. Nonparametric tests (Kruskal-Wallis tests) were used to assess the associations between items in the blended learning readiness questionnaires and the medical educators' demographic characteristics, teaching roles, and training experience. The results of the statistical analyses are presented as the means ± SD or medians, depending on response distribution, and are considered statistically significant when $P < 0.05$.

## RESULTS

### Representativeness of survey data regarding anatomists' acceptance of blended learning

The survey was distributed to a WeChat group consisting of 496 members; the questionnaire was completed by a total of 297 anatomy teachers, their responses were included for subsequent analysis. Consequently, the achieved response rate for the survey was 59.88%. These respondents were geographically distributed across mainland China, rendering their responses representative of Chinese anatomy educators. The respondents'

**Table 1  Demographic characteristics of the participants in this study.**

| Variables | Values *n* (%) |
|---|---|
| **Gender** | |
| *Male* | 167 (56.2%) |
| *Female* | 130 (43.8%) |
| **Age** | |
| *21–30* | 13 (4.4%) |
| *31–40* | 81 (27.3%) |
| *41–50* | 131 (44.1%) |
| *51–60* | 66 (22.2%) |
| *>60* | 6 (2.0%) |
| **Teaching years** | |
| *<10* | 72 (24.2%) |
| *11–20* | 113 (38.1%) |
| *21–30* | 73 (24.6%) |
| *>30* | 39 (13.1%) |
| **Job titles** | |
| *Assistant professor* | 18 (6.1%) |
| *Lecturer* | 84 (28.3%) |
| *Associate professor* | 97 (32.6%) |
| *Professor* | 98 (33.0%) |
| **Professional training on education** | |
| *Formal* | 237 (79.8%) |
| *Informal* | 56 (18.9%) |
| *Not confirmed* | 4 (1.3%) |
| **Support from mentor** | |
| *Yes* | 230 (56.2%) |
| *No* | 67 (56.2%) |
| **Support from in-service institutes** | |
| *Low* | 34 (11.4%) |
| *Medium* | 125 (42.1%) |
| *High* | 138 (46.5%) |

Notes.
*n*, number.
$n = 297/100\%$.

demographic information was summarized in Table 1. The anatomists who participated exhibited several distinct characteristics: a slightly higher proportion of male anatomists ($n = 167/56.2\%$) compared to female anatomists ($n = 130/43.8\%$); a majority of the anatomists ($n = 131/44.1\%$) were aged between 41 and 50; and most of them had extensive teaching experience, with an average of $17.72 \pm 9.62$ years in the field of anatomy education. Nearly 80% of respondents reported having received formal teaching training, and at the outset of their careers, they benefited mentor guidance. The Cronbach's $\alpha$ of the 34 items indicating the anatomy teachers' attitudes toward blended learning was 0.93, showing a high reliability of the survey instrument.

## Priorities of anatomists for blended learning: flexibility, community, and management

First, the responses of the anatomists about their attitudes toward blended learning revealed a strong emphasis on learning flexibility, as indicated by the highest scores assigned to Item 1: "unlimited access to lecture materials", along with Items 2 and 4: "can choose where and when to study", and Item 3: "study at one's own pace" (Table 2). The second-highest scores were attributed to items related to connected learning with a community-centered learning environment. These items encompassed Item 26: "study better *via* classroom activities", Item 27: "study better when being guided personally", Item 24: "prefer to receive feedback quickly in classroom lectures", Item 25: "study more effectively when collaborating with others in the classroom". Furthermore, respondents did not resist online learning (Item 7), and exhibited a favorable disposition toward related technologies: "I believe the Web is a useful platform for learning" (Item 19), and "I think we should use technologies in learning" (Item 22). The distribution of responses skewed leftward (skewness <0), indicating the generally positive stance of surveyed educators toward the questionnaire statements (Table 2).

Next, six factors encapsulating anatomists' attitudes toward blended learning were identified by the principal factor analysis: connected learning (14.77%), learning control (13.50%), learning flexibility (12.23%), online interaction (11.76%), mastery of related technology (9.65%) and negative attitude toward online learning (7.31%, Table 3). These factors reflected the most distinctive aspects of surveyed respondents' perceptions. Notably, certain questionnaire responses showed relatively negative attitudes toward online learning and ranked lower in terms of reliability among the principal factors. Although all six factors collectively accounted for 69.2% of the variance, they comprised an effective index for discerning noticeable variations in the perceptions of surveyed anatomists.

Descriptive analysis based on the identified factors reinforced the statistical characteristics of the factors (Table 4), with learning flexibility ($n = 6$, $5.05 \pm 0.90$), connected learning ($n = 7$, $4.46 \pm 0.63$), mastery of online learning technology ($n = 3$, $4.39 \pm 0.99$), and learning management ($n = 6$, $4.32 \pm 1.00$), leading the rankings in descending order.

## Association of anatomists' blended learning preferences with institutional support and job titles

Nonparametric analysis was performed to explore demographic characteristics and professional experiences that might influence anatomists' perceptions of blended learning. The findings demonstrated minimal influence from age, gender, and years of experience on anatomists' attitudes. However, statistically significant associations emerged between job titles, support from mentors and institutions, and attitudes toward blended learning. In particular, significant differences in responses were observed when grouped by levels of support (Table 5).

Given the importance of institutional support, an exploration into its underlying mechanism was conducted. Firstly, support levels were re-coded as low (Likert scales =1 and 2), medium (Likert scales =3 and 4), and high (Likert scales =5 and 6). The results

**Table 2** Descriptive characteristics of the anatomists' attitudes towards blended learning.

| Questions | Median | IQR | | Skewness |
|---|---|---|---|---|
| | | Q¹ | Q³ | |
| 1. I would like unlimited access to lecture materials. | 6 | 5 | 6 | −1.52 |
| 2. I would like to decide where I want to study. | 6 | 5 | 6 | −1.38 |
| 3. I like to study at my own pace. | 6 | 4 | 6 | −1.26 |
| 4. I would like to decide when I want to study. | 5 | 4 | 6 | −1.27 |
| 5. I believe face-to-face learning is more effective than online learning. | 5 | 4 | 6 | −1.13 |
| 6. I am comfortable with self-directed learning. | 5 | 4 | 6 | −0.85 |
| 7. I do not resist having my lessons online. | 5 | 4 | 6 | −0.97 |
| 8. I like online learning as it provides richer instructional content. | 5 | 4 | 6 | −0.78 |
| 15. I can study over and over again online. | 5 | 4 | 6 | −0.98 |
| 19. I believe the Web is a useful platform for learning. | 5 | 4 | 6 | −0.77 |
| 22. I think we should use technologies in learning. | 5 | 4 | 6 | −0.67 |
| 23. I have a sense of community when I meet other students in the classroom. | 5 | 4 | 6 | −0.68 |
| 24. I like the fast feedback when I meet my lecturer in person. | 5 | 4 | 6 | −0.99 |
| 25. I find learning through collaboration with others face-to-face is more effective. | 5 | 4 | 6 | −0.94 |
| 26. I learn better through lecturer-directed classroom-based activities. | 5 | 4 | 6 | −1.01 |
| 27. I learn better when someone guides me personally. | 5 | 4 | 6 | −1.09 |
| 29. I am comfortable in using Web technologies to exchange knowledge with others. | 5 | 4 | 6 | −0.43 |
| 30. I would like to interact with my lecturer online. | 5 | 4 | 6 | −0.47 |
| 31. I would like to interact with other students outside of the classroom. | 5 | 4 | 5 | −0.56 |
| 9. I would like lecture time in the classroom to be reduced. | 4 | 3 | 5 | −0.26 |
| 11. I get bored when studying online. | 4 | 2 | 4 | −0.08 |
| 13. I am more likely to miss assignment due dates in an online learning environment. | 4 | 3 | 4.5 | −0.11 |
| 14. I organize my time better when studying online. | 4 | 3 | 5 | −0.19 |
| 16. Online learning motivates me to prepare well for my studies. | 4 | 3 | 5 | −0.27 |
| 17. Online learning encourages me to make plans. | 4 | 3 | 5 | −0.26 |
| 18. Online learning makes me more responsible for my studies. | 4 | 3 | 5 | −0.19 |
| 20. I am familiar with Web technologies. | 4 | 3 | 5 | −0.21 |
| 21. I find Web technologies easy to use. | 4 | 4 | 5 | −0.35 |
| 28. I feel isolated in an online learning environment. | 4 | 3 | 5 | −0.22 |
| 32. I find it easy to communicate with others online. | 4 | 4 | 5 | −0.21 |
| 33. I appreciate easy online access to my lecturer. | 4 | 4 | 5 | −0.42 |
| 34. I can collaborate well with a virtual team in doing assignments. | 4 | 3 | 5 | −0.34 |
| 10. I would like to have my classes online rather than in the classroom. | 3 | 2 | 4 | 0.26 |
| 12. I find it very difficult to study online. | 3 | 2 | 4 | 0.31 |

**Notes.**
The survey data is gained from total 297 anatomists ($n = 297$).
IQR, interquartile range; Q¹, quartile at the 25th; Q³, quartile at the 75th.
Likert scales are 1-6 for the questionnaire items.

of Kruskal–Wallis tests indicated significant differences in anatomists' attitudes across the various levels of support. Specific aspects positively linked to high institutional support

**Table 3  Summary of principal factor analysis of the questionnaire answered by the anatomists about their attitudes towards blended learning.**

| Questions | Factor 1 | Factor 2 | Factor 3 | Factor 4 | Factor 5 | Factor 6 |
|---|---|---|---|---|---|---|
| 24. I like the fast feedback when I meet my lecturer in person. | 0.85 | | | | | |
| 26. I learn better through lecturer-directed classroom-based activities. | 0.83 | | | | | |
| 25. I find learning through collaboration with others face-to-face is more effective. | 0.82 | | | | | |
| 27. I learn better when someone guides me personally. | 0.81 | | | | | |
| 5. I believe face-to-face learning is more effective than online learning. | 0.57 | | | | | |
| 23. I have a sense of community when I meet other students in the classroom. | 0.54 | | | | | |
| 31. I would like to interact with other students outside of the classroom. | 0.46 | | | | | |
| 17. Online learning encourages me to make plans. | | 0.85 | | | | |
| 16. Online learning motivates me to prepare well for my studies. | | 0.84 | | | | |
| 18. Online learning makes me more responsible for my studies. | | 0.81 | | | | |
| 14. I organize my time better when studying online. | | 0.64 | | | | |
| 15. I can study over and over again online. | | 0.61 | | | | |
| 19. I believe the Web is a useful platform for learning. | | 0.48 | | | | |
| 3. I like to study at my own pace. | | | 0.85 | | | |
| 4. I would like to decide when I want to study. | | | 0.85 | | | |
| 2. I would like to decide where I want to study. | | | 0.83 | | | |
| 1. I would like unlimited access to lecture materials. | | | 0.64 | | | |
| 6. I am comfortable with self-directed learning. | | | 0.56 | | | |
| 7. I do not resist having my lessons online. | | | 0.47 | | | |
| 9. I would like lecture time in the classroom to be reduced. | | | | 0.68 | | |
| 33. I appreciate easy online access to my lecturer. | | | | 0.64 | | |
| 30. I would like to interact with my lecturer online. | | | | 0.61 | | |
| 34. I can collaborate well with a virtual team in doing assignments. | | | | 0.61 | | |
| 32. I find it easy to communicate with others online. | | | | 0.61 | | |
| 10. I would like to have my classes online rather than in the classroom. | | | | 0.583 | | |
| 29. I am comfortable in using Web technologies to exchange knowledge with others. | | | | 0.57 | | |
| 8. I like online learning as it provides richer instructional content. | | | | 0.44 | | |
| 20. I am familiar with Web technologies. | | | | | 0.81 | |
| 21. I find Web technologies easy to use. | | | | | 0.79 | |
| 22. I think we should use technologies in learning. | | | | | 0.50 | |

**Table 3** (*continued*)

| Questions | Factor 1 | Factor 2 | Factor 3 | Factor 4 | Factor 5 | Factor 6 |
|---|---|---|---|---|---|---|
| 12. I find it very difficult to study online. | | | | | | 0.87 |
| 11. I get bored when studying online. | | | | | | 0.76 |
| 13. I am more likely to miss assignment due dates in an online learning environment. | | | | | | 0.76 |
| 28. I feel isolated in an online learning environment. | | | | | | 0.50 |
| **Reliability** | 0.72 | 0.91 | 0.90 | 0.88 | 0.84 | 0.75 |
| **% of Variance** | 14.77 | 13.50 | 12.23 | 11.76 | 9.65 | 7.31 |
| **Key Factors** | Connected learning | Learning control | Learning flexibility | Online interaction | Mastery of related technology | Negative attitude towards online learning |

**Notes.**

Extraction methods: Principal Component Analysis. Rotation methods: Varimax Kaiser normalization (KMO = 0.92). Rotation converged in 31 iterations. $n = 297$. The number of factors were determined by the eigenvalues extracted greater than 1. Reliability is the Cronbach $\alpha$ of each factor, "% of the variance" is the percentage of the variance that the factor can explain of the data set.

**Table 4  Descriptive data of the identified factors.**

| Factors | $n$ | Average sum value | Mean | SD | Median | Skewness |
|---|---|---|---|---|---|---|
| 1. Connected learning | 7 | 31.21 | 4.46 | 0.63 | 5 | −0.93 |
| 2. Learning control | 6 | 25.92 | 4.32 | 1.00 | 4 | −0.47 |
| 3. Learning flexibility | 6 | 33.53 | 5.05 | 0.90 | 5 | −1.28 |
| 4. Online interaction | 8 | 33.53 | 4.19 | 0.94 | 4 | −0.28 |
| 5. Mastery of the on-line learning related technology | 3 | 13.17 | 4.39 | 0.99 | 4 | −0.32 |
| 6. Negative attitude to-wards online learning | 4 | 14.41 | 3.60 | 1.08 | 4 | −0.07 |

**Notes.**

$n$, numbers of the related items of the survey;  Average sum value,  average sum value of the factor;  Mean,  mean of the total items of the factor;  Median,  the middle value of the answers to total items of the factor.

Likert scales are 1–6 for the questionnaire items.

encompassed a stronger inclination toward online learning (Table 5), better learning control (Items 16–18), and adeptness in internet technology(Items 19–21). Moreover, those anatomists preferring interacting with others were more likely to receive institutional support, fostering an appropriate learning community environment (Items 23, 26, and 29). Lecturers displayed notable engagement in the connected learning community when divided by job titles. They preferred to interact with mentors, colleagues, and students both online and offline, demonstrating robust communicative skills and easy mentor connections. Anatomists mentored by experienced professionals exhibited heightened awareness of learning control. Lastly, correlation between factors derived from anatomists' attitudes and job titles, mentor guidance, and institutional support were analyzed (Table 6). Statistically significant differences were more pronounced when considering different levels of institutional support. Aspects significantly tied to high institutional support included increased learning flexibility, improved learning control, and mastery of online learning

Cheng et al. (2023), *PeerJ*, DOI 10.7717/peerj.16283

**Table 5  Descriptive statistics and correlative analysis of the anatomists' attitudes toward blended learning significantly related to job titles, guidance from mentors and the different levels of support from the institutes.**

| Questions | Job titles | | | | | | Guidance from mentors | | | | Supports from institutes | | | | |
| --- | --- | --- | --- | --- | --- | --- | --- | --- | --- | --- | --- | --- | --- | --- | --- |
| | Assistant professor $n = 18$ | Lecturer $n = 84$ | Associate professor $n = 97$ | Professor $n = 98$ | H | P | Yes $n = 230$ | No $n = 67$ | H | P | Low $n = 34$ | Medium $n = 125$ | High $n = 138$ | H | P |
| | | | | | | df = 3 | | | | df = 1 | | | | | df = 2 |
| 6. I am comfortable with self-directed learning. | 4 | 5 | 5 | 5 | 1.04 | 0.79 | 5 | 5 | 1.34 | 0.25 | 4.5 | 5 | 5 | 10.72 | **0.01** |
| 7. I do not resist having my lessons online. | 5.5 | 5 | 5 | 5 | 2.03 | 0.57 | 5 | 5 | 2.34 | 0.13 | 4.5 | 5 | 5 | 13.52 | **< 0.01** |
| 8. I like online learning as it provides richer instructional content. | 4.5 | 5 | 5 | 5 | 2.45 | 0.48 | 5 | 4 | 2.72 | 0.10 | 4 | 4 | 5 | 11.27 | **< 0.01** |
| 16. Online learning motivates me to prepare well for my studies. | 4 | 5 | 4 | 4 | 5.14 | 0.16 | 4 | 4 | 3.98 | **0.05** | 4 | 4 | 4 | 7.50 | **0.02** |
| 17. Online learning encourages me to make plans. | 4 | 4 | 4 | 4 | 2.13 | 0.55 | 4 | 4 | 5.47 | **0.02** | 4 | 4 | 4 | 6.06 | **0.05** |
| 18. Online learning makes me more responsible for my studies. | 4 | 4 | 4 | 4 | 0.33 | 0.96 | 4 | 4 | 3.14 | 0.08 | 4 | 4 | 4 | 6.95 | **0.03** |
| 19. I believe the Web is a useful platform for learning. | 5 | 5 | 5 | 5 | 0.64 | 0.89 | 5 | 5 | 1.91 | 0.17 | 5 | 5 | 5 | 7.72 | **0.02** |
| 20. I am familiar with Web technologies. | 4 | 4 | 4 | 4 | 6.54 | 0.09 | 4 | 4 | 0.78 | 0.38 | 4 | 4 | 4 | 10.03 | **0.01** |
| 21. I find Web technologies easy to use. | 4 | 4 | 4 | 4 | 4.69 | 0.20 | 4 | 4 | 1.56 | 0.21 | 4 | 4 | 4 | 6.86 | **0.03** |
| 23. I have a sense of community when I meet other students in the classroom. | 4.5 | 5 | 5 | 5 | 1.83 | 0.61 | 5 | 5 | 3.36 | 0.07 | 4 | 5 | 5 | 6.74 | **0.03** |
| 26. I learn better through lecture-directed classroom-based activities. | 5 | 5 | 5 | 5 | 1.63 | 0.65 | 5 | 5 | 0.01 | 0.93 | 5 | 5 | 5 | 6.54 | **0.04** |
| 29. I am comfortable in using Web technologies to exchange knowledge with others. | 4 | 5 | 5 | 5 | 1.54 | 0.67 | 5 | 4 | 1.39 | 0.24 | 5 | 4 | 5 | 10.63 | **0.01** |

*(continued on next page)*

**Table 5** (*continued*)

| Questions | Job titles | | | | | | Guidance from mentors | | | | Supports from institutes | | | | |
|---|---|---|---|---|---|---|---|---|---|---|---|---|---|---|---|
| | Assistant professor $n = 18$ | Lecturer $n = 84$ | Associate professor $n = 97$ | Professor $n = 98$ | $H$ | $P$ | Yes $n = 230$ | No $n = 67$ | $H$ | $P$ | Low $n = 34$ | Medium $n = 125$ | High $n = 138$ | $H$ | $P$ |
| | | | | | $df = 3$ | | | | $df = 1$ | | | | | $df = 2$ | |
| 30. I would like to interact with my lecturer online. | 4 | 5 | 5 | 4 | 12.15 | **0.01** | 5 | 4 | 1.44 | 0.23 | 5 | 4 | 5 | 4.60 | 0.10 |
| 31. I would like to interact with other students outside of the classroom. | 4 | 5 | 5 | 4 | 9.56 | **0.02** | 5 | 5 | 0.07 | 0.79 | 5 | 4 | 5 | 2.82 | 0.24 |
| 32. I find it easy to communicate with others online. | 4 | 5 | 4 | 4 | 9.62 | **0.02** | 4 | 4 | 1.79 | 0.18 | 4 | 4 | 4 | 5.75 | 0.06 |
| 33. I appreciate easy online access to my lecturer. | 4 | 5 | 4 | 4 | 15.89 | **0.01** | 4 | 4 | 4.70 | **0.03** | 4 | 4 | 4.5 | 2.30 | 0.32 |

**Notes.**

The data reported are the medians of the item in each group, using nonparametric method (*H: H* values of Kruskal-Wallis H tests). The bold values show the statistical significant difference.

technology (Table 6). Notably, negative attitude toward online learning remained consistent across different job titles, mentor guidance, and institutional support levels.

## DISCUSSION

The impact of the COVID-19 pandemic has brought forth heightened challenges in education, including faculty development (*Ahmed, Allaf & Elghazaly, 2020*; *Gallagher & Schleyer, 2020*; *Rose, 2020*). Swift shifts and abrupt alterations in teaching and assessment modalities have underscored the urgent need for medical educators to enhance their initial pedagogical competencies in online and blended learning, often surpassing their preparedness. This juncture demands introspection, collaborative learning, and continuous adaption to the evolving landscape. In this regard, faculty development is pivotal in assisting educators through uncertainty and embracing change, facilitating the seamless integration of educational curricula onto online platforms, and promoting more effective education for prospective health professionals (*Steinert, Irby & Dolmans, 2021*).

The necessity for professional training in online and blended learning is especially pressing for anatomists working in innovative teaching environments. A parallel study has disclosed a shift from predominantly face-to-face teaching to a blended learning format has occurred in anatomy education across China since the onset of the COVID-19 pandemic. Blended learning emerges as an adaptable solution for educational institutions confronted with sudden transformation and striving for sustainable progress. Moreover, the majority of medical schools in mainland China have prioritized faculty professional training in online and blended learning (*Cheng et al., 2023*). Evaluating the effectiveness of faculty training and development can be achieved by soliciting input from faculty members, as their self-perceived usefulness encourages active engagement in subsequent educational practices. Furthermore, learning is contextually dependent and necessitates appropriate opportunities for the application of newly acquired knowledge, enabling immediate real-world practice. Therefore, the effectiveness of training is primarily associated with the anatomists' perspectives on and attitudes toward online and blended learning training.

Chinese anatomists highly valued learning flexibility as the most critical factor (Tables 2 and 4). The superiority of an online and blended learning framework resides in its ability to provide learners with unlimited access to learning material and facilitate global communication between instructors and learners through web-based technology (*Naidu, 2019*). This feature endows learners with exceptional convenience and the freedom to study without limitation. As educators, the Chinese anatomists viewed online learning not merely as a supplementary component to traditional teaching but as a valuable entity in its own right. This finding bolsters our confidence in shaping a more open and adaptable online and blended learning environment for future teaching endeavors (*Oliver, 1999*). However, despite learning flexibility garnered significant value, the most important factor identified by anatomists was connected learning (Table 3). This underscores their eagerness to foster an atmosphere of community-based learning. Such an environment involves purposeful connection between instructors and learners in either a physical classroom or a virtual learning environment (instructional link), learners' active participation in panel discussions

**Table 6 Descriptive statistics of the factors related to job titles, guidance from mentors and the different levels of support received.**

| Questions | Job titles | | | | | | Guidance from mentors | | | | Supports from institutes | | | | |
|---|---|---|---|---|---|---|---|---|---|---|---|---|---|---|---|
| | Assistant professor n = 18 | Lecturer n = 84 | Associate professor n = 97 | Professor n = 98 | *H* | *P* | Yes n = 230 | No n = 67 | *H* | *P* | Low n = 34 | Medium n = 125 | High n = 138 | *H* | *P* |
| | | | | | *df* = 3 | | | | *df* = 1 | | | | | *df* = 2 | |
| 7. Connected learning | 4 | 5 | 5 | 5 | 5.01 | 0.17 | 5 | 5 | 0.54 | 0.46 | 5 | 5 | 5 | 3.35 | 0.19 |
| 8. Learning control | 4 | 5 | 4 | 4 | 2.12 | 0.55 | 4 | 4 | 2.57 | 0.11 | 4 | 4 | 5 | 7.12 | **0.03** |
| 9. Learning flexibility | 5 | 6 | 5 | 5 | 1.33 | 0.72 | 5 | 5 | 0.00 | 0.98 | 5 | 5 | 6 | 10.27 | **0.01** |
| 10. Online interaction | 4 | 4 | 4 | 4 | 8.10 | **0.04** | 4 | 4 | 5.19 | **0.02** | 4 | 4 | 4 | 5.56 | 0.06 |
| 11. Mastery of the on-line learning-related technology | 4 | 5 | 4 | 4 | 6.44 | 0.09 | 4 | 4 | 0.81 | 0.37 | 4 | 4 | 5 | 9.51 | **0.01** |
| 12. Negative attitude toward online learning | 3 | 3 | 4 | 4 | 3.00 | 0.39 | 4 | 3.5 | 0.00 | 0.95 | 4 | 3 | 4 | 0.56 | 0.76 |

**Notes.**
The data reported are medians of the item in each group, using nonparametric method (Kruskal-Wallis H tests). The bold values show the statistically significant difference.

or teamwork under instructors' guidance (community integration), and learners' active engagement in research projects in collaboration with researchers (community participation) (*Zhu & Baylen, 2005*). This preference for community-based learning stems from its effectiveness in promoting outcome-based education, encouraging peer-to-peer learner interactions, and providing firsthand experiences through pedagogically oriented activities (*Chang, 2012*).

The anatomists in this study also accorded significance to learning management in the context of online and blended learning. Learning management encompasses adept self-control abilities, and are vital for achieving better outcomes in blended learning. This result reminds the importance of designing pedagogical approaches that foster intrinsic motivation and lead to improved learning outcomes for learners (*Weaver, Spratt & Nair, 2008*). The results also suggested that anatomists who effectively managed online technology and held positive perceptions toward online and blended learning are more likely to assume active and affirmative roles in delivering such learning experiences (Table 4). This finding echoes a previous report which highlighted the positive impact of structured faculty training programs, leading to high participant satisfaction, positive shift in teaching attitudes, increased knowledge and skills, and noticeable transformation in teaching behaviors (*Steinert et al., 2016*). These results align with the Kirkpatrick levels of educational outcomes, emphasizing the importance of strengthening healthcare professional training to amplify teaching effectiveness and positively impact students (*Piryani et al., 2018*; *Steinert et al., 2006*).

Further exploration is imperative to understand the key driving forces influencing Chinese anatomists' perspectives toward blended learning. This understanding holds significance for shaping administrative policies and tailoring anatomist-specific training for faculty development at medical schools. The primary factor influencing anatomists' perspectives toward online and blended learning is the support they garner from mentors and in-service medical schools, along with their designated job titles (Table 5). Delving deeper into the reasons underlying this predominant influence revealed the pivotal role played by anatomists' self-management abilities and features. In essence, as Chinese anatomists became more adept at blended learning, they are better poised to tap into various forms of support, leading to improved learning outcomes. Another pivotal source of support was the acknowledgement of the learning community, where anatomists with an interest in education or collaborative study converge (*DuFour, 2004*; *Shea, 2006*). These findings indicated that our medical schools and administration should bolster diverse forms of support, to ensure faculty members' dedication and commitment to educational initiatives. In addition to institutional support, this study highlights the value of mentorship from supportive co-teachers in catalyzing faculty development (*Jackevicius et al., 2014*; *Vitale, 2010*). Teaching competencies are largely cultivated through observation, co-teaching experience, and feedback from senior teachers, fostering reflective observation. However, such a process is not easily attained through online learning, elucidating why anatomy teachers emphasized the significance of connected learning. The results suggest that establishing an authentic or virtual connected learning community could offer a

sustainable approach for faculty training programs, optimizing the effectiveness of faculty development.

A notable facet that cannot be overlooked is the presence of negative attitudes toward online learning, ranking sixth among the factors related to blended learning. These comments suggest that face-to-face education remains irreplaceable in medical education, especially for anatomy education, which benefits from hands-on laboratory modalities, delivery formats, and assessments. These aspects provide indispensable benchmarks for determining the most effective anatomical practice for continuous development. Unquestionably, face-to-face instruction maintains a fundamentally stable teaching format.

## LIMITATIONS

Several limitations must be acknowledged when interpreting the survey and its results. Firstly, the survey results may not be generalizable to all anatomists in China due to the potential influence of selection bias. Secondly, the survey was conducted using a cross-sectional method. Although we previously performed a study among Chinese anatomy educators about online teaching in April 2020 (*Cheng et al., 2021*), it remains challenging to longitudinally assess anatomists' attitudes toward online and blended learning over the two years following the COVID-19 pandemic. The difficulty arised from the non-identical nature of the questions used in these two surveys. Thirdly, the survey data were analyzed quantitatively, and thus, an analytical interpretation of qualitative data, such as focus groups, was absent from this study. Qualitative analysis could have provided a comprehensive understanding of the experiences of anatomy teachers regarding the "support" received at universities in China. Lastly, the survey did not delve into the specific blended learning models adopted by anatomists, including the resources and platforms used or the amount of time dedicated to blended teaching activities. Inclusion of these details could have provided valuable insights into the practical aspects of blended learning in anatomy education.

## CONCLUSION

Over the past two years, the COVID-19 pandemic has greatly impacted and altered faculty development, especially in training programs related to online and blended learning. Findings from this survey conducted among anatomists across mainland China shed light on the perspectives of Chinese anatomists regarding online and blended learning. The findings underscore that anatomists perceive learning flexibility, the presence of a connected learning community, and effective learning management as pivotal features of online and blended learning. Notably, support from mentors and institutions emerges as a significant factor shaping positive attitudes toward online and blended learning, subsequently correlating with improved training outcomes. The specific forms of support

required may vary based on the anatomists' proficiency in learning management abilities and the particular features of the learning milieu.

### Funding
This work was supported by the Research Projects of Pedagogical Reform at Jinan University (JG2022127), Research Projects supported by the MOOC Committee of Undergraduates Courses in Guangdong Province (2022ZXKC035), and the Medical Teaching and Educational Management Reform Research Project Foundation of Jinan University (21YXJG043). The funders had no role in study design, data collection and analysis, decision to publish, or preparation of the manuscript.

### Grant Disclosures
The following grant information was disclosed by the authors:
Research Projects of Pedagogical Reform at Jinan University: JG2022127.
MOOC Committee of Undergraduates Courses in Guangdong Province: 2022ZXKC035.
Medical Teaching and Educational Management Reform Research Project Foundation of Jinan University: 21YXJG043.

### Competing Interests
The authors declare there are no competing interests.

### Author Contributions
- Xin Cheng conceived and designed the experiments, performed the experiments, analyzed the data, prepared figures and/or tables, authored or reviewed drafts of the article, and approved the final draft.
- Jian Bai analyzed the data, prepared figures and/or tables, and approved the final draft.
- San-Qiang Pan analyzed the data, prepared figures and/or tables, and approved the final draft.
- Yun-Qing Li conceived and designed the experiments, performed the experiments, authored or reviewed drafts of the article, and approved the final draft.
- Xuesong Yang conceived and designed the experiments, authored or reviewed drafts of the article, and approved the final draft.

### Human Ethics
The following information was supplied relating to ethical approvals (*i.e.*, approving body and any reference numbers):

The study was conducted with ethics approval from the Research Ethics Committee of Jinan University (No. JNUKY-2021-038).

### Data Availability
The raw data are available in the Supplemental File.
## Supplemental Information

Supplemental information for this article can be found online at http://dx.doi.org/10.7717/peerj.16283#supplemental-information.

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
