# Peer review of "Assessing Chinese anatomists' perceptions and attitudes toward blended learning through faculty development training programs"

_PeerJ, doi:10.7717/peerj.16283_

## Round 0.1 · original submission · Major Revisions

I feel that the first two reviewers have valid points. The more critical of the reviewers. reviewer number 1 has provided clear direction to progress the manuscript forward for further review, please pay careful attention to this reviewer's suggestions.

Reviewer 1 ·

Basic reporting

Thank you for inviting me to review this article.
I found this manuscript a little difficult to read. I strongly recommend the authors seek professional language editing services before resubmission.
The structure of the article is acceptable, though I would suggest the authors shorten the introduction section and focus on the problem (why did you need to do this study), the gap (what was known and what needs to be explored), and the hook (what would you do with the results, what would be your next step).

Experimental design

I have some concerns about the methods for this study.
(1) What is the population of interest in this study? The link to the survey was distributed to the group of anatomy department directors. So were they the population of interest? Was the link to the survey forwarded to other anatomy staff by the department directors? These questions need to be clarified.
(2) I would be reluctant to call it a "nationwide survey" because the sampling frame is from a chatting group, which definitely does not cover the population of interest.
(3) Mean rank doesn't have any clinical implications. Please consider reporting the data with medians (IQR) if the data are highly skewed.
(4) Although the author briefly addressed the development of the survey and pilot test process, it would be helpful if some validity and reliability data of the original survey be reported.

Validity of the findings

The results section will need some work as well.
(1) How many surveys were totally distributed? What was the response rate?
(2) Table 5 and Table 6 can be combined. Please consider presenting medians (IQR) instead of mean ranks
(3) Please provide exact p-values for table 7 and use medians (IQR) instead of mean ranks.

Additional comments

I would consider the following points for discussion.
(1) Why is this study unique? How are Chinese anatomy education and Chinese anatomy teachers different from the rest of the world?
(2) what are the educational implication of the results? Or what would you do with the results of this study? What are the future directions? Especially when the pandemic subsides and face-to-face teaching becomes possible, what are the implications of the results?

Reviewer 2 ·

Basic reporting

The authors have addressed an interesting topic regarding the necessity of faculty development using online/blended learning approach. The topic this study tries to touch on is not so often reported but require much attention, perhaps with a need for more attention than ever, as the COVID hits with challenges as well as other technological advancement in teaching and learning.

The language of this manuscript is friendly for international readers and easy to understand while made clear about delivering their ideas regarding the topic. The tables have also provided critical information

One thing that should be noted is that the Limitation part should be placed before the Conclusion (as the last part of the Discussion).

Experimental design

This is an observational descriptive study that uses survey to discover the perceptions and attitudes toward blended learning through faculty development training programs among Chinese anatomists,
The overall design is fair adapted from previous reported instrument, which provides certain validity evidence. Convenience sampling do weaken the reliability of the results, but in this case, I do think is acceptable, given no prior research could provide enough information on the sampling N number.

Validity of the findings

The authors have drawn interesting conclusions from the survey, which could serve as potential advices for improving and expanding technology-enhanced anatomical teaching and learning.

Reviewer 3 ·

Basic reporting

Generally speaking, this manuscript was clearly written, and the flow of the background, justification, experiment, results, and conclusion sections were straightforward and the arguments proffered made logical sense. The need for this research, background context, and the scientific basis of the work using references from prior literature were thorough. Furthermore, the tables were laid out well and were formatted correctly.

Experimental design

The research design laid forth in this manuscript was simple. This work relied on a web-based survey given to Chinese instructor anatomists. Basic demographic information was collected, and key questions were asked regarding their beliefs concerning classroom-based, virtual, and blended instruction. Researchers were keenly interested in understanding instructor perceptions of blended learning, due to the fact that the pandemic has rapidly forced academic settings to pivot traditional learning methodologies.

The methods described were appropriate and the statistical analyses were complete. Given that this was only a survey, the goal of the work was to simply understand perceptions of Chinese instructor anatomists.

Validity of the findings

The researchers utilized relevant statistical analyses to quantitatively analyze the qualitative results coming out of their survey. They classified the results in an appropriate way by looking at factors, and were able to account for a high degree of variance with their methods. The impact of the results was that blending learning possesses some highly valuable features, including access to relevant content and materials anytime and anywhere, but the need for collaboration and hands-on opportunities to learn is still highly desired. As such, conclusions were well-articulated on the basis of what was examined by the survey.

Additional comments

The survey examined Chinese instructor anatomist perceptions of blended learning at the general level. There were no specific learning materials that were being assessed or critiqued, and no specific hypotheses to test. I suspect there would be high variability in the quality of learning materials that may check the boxes for what was being surveyed, that would lead to vastly different levels of acceptance due to their perceived value.

Although understanding perceptions and openness to these methods is of high importance to the community, and the writing, experiment, analysis, and conclusions were all appropriate in this work, my only qualm with the work is that I am not sure if publication of survey data alone is sufficient for inclusion as a full research article. That being said, whether or not survey data alone is enough is an Editor's decision. If survey data is sufficient, then I would definitely accept the article.

---

## Round 0.2 · Minor Revisions

Please see the very minor revision requests as it pertains to the current iteration of your manuscript. Thank you for your continued process in the revision process.

Reviewer 1 ·

Basic reporting

Thank you for inviting me to review this manuscript. I'm glad to see the manuscript improved significantly especially the structure of the introduction and discussion. I still have a few minor concerns re: methods and results.

Note: All line numbers refer to the track change document (PDF), not the clean version.

Line 170: Please mention that the data were described as mean (SD) or median (IQR) depending on the distribution of the response.

Line 186 - 188 : "Next, the anatomy teachers....perspectives as leaners" This sentence seems to describe the methods and should not appear in the result section.

Line 179, Table 1: I struggled to understand the wording of "control variables". I suggest the authors revise it as "demographic characteristics of participants".

In the response to the reviewers, the author indicates the response rate of the survey was 59.88%. Please report this in the result section of the manuscript.

Table 2: I would remove the SD column, because SD is usually used when data were normally distributed. I would also remove Q2 column because it was the same as median by definition. I would replace quartiles with IQR (Q1 - Q3).

Experimental design

No coment

Validity of the findings

No comment

Additional comments

No comment

---

## Round 0.3 · Minor Revisions

Thank you for resubmitting your manuscript and accepting the various reviewers recommendations. Please review your manuscript for English Language nuance, style and consistency.

**Language Note:** The Academic Editor has identified that the English language must be improved. PeerJ can provide language editing services - please contact us at copyediting@peerj.com for pricing (be sure to provide your manuscript number and title). Alternatively, you should make your own arrangements to improve the language quality and provide details in your response letter. – PeerJ Staff

---

## Round 0.4 · accepted · Accept

Thank you for your patience and multiple revisions as it pertains to your manuscript. The most recent review confirms requested revisions.

Reviewer 2 ·

Basic reporting

The authors had made extensive revision regarding the language issue. The current version is more friendly to read.

Experimental design

no comment

Validity of the findings

no comment

Additional comments

no comment